# Development of Novel Pasta Products with Evidence Based Impacts on Health—A Review

**DOI:** 10.3390/foods11010123

**Published:** 2022-01-04

**Authors:** Mike Sissons

**Affiliations:** Department of Primary Industries, Tamworth Agricultural Institute, 4 Marsden Park Road, Tamworth, NSW 2340, Australia; mike.sissons@dpi.nsw.gov.au

**Keywords:** durum wheat, pasta, health benefits, functional pasta, functional food

## Abstract

Pasta made from durum wheat is a widely consumed worldwide and is a healthy and convenient food. In the last two decades, there has been much research effort into improving the nutritional value of pasta by inclusion of nonconventional ingredients due to the demand by health-conscious consumers for functional foods. These ingredients can affect the technological properties of the pasta, but their health impacts are not always measured rather inferred. This review provides an overview of pasta made from durum wheat where the semolina is substituted in part with a range of ingredients (barley fractions, dietary fibre sources, fish ingredients, herbs, inulin, resistant starches, legumes, vegetables and protein extracts). Impacts on pasta technological properties and in vitro measures of phytonutrient enhancement or changes to starch digestion are included. Emphasis is on the literature that provides clinical or animal trial data on the health benefits of the functional pasta.

## 1. Introduction

Carbohydrates in foods are an important source of energy for humans, with cereals, tubers and pulses being the main dietary sources. Pasta is a popular food worldwide known for its ease of preparation, good storage stability (dried form), low cost, simple preparation with a low glycaemic index (GI). Pasta consists mostly of carbohydrates (70–76%), protein (~10–14%), lipids (~1.8%), dietary fibre (~2.9%) and small amounts of minerals and vitamins [1]. Pasta is made from either semolina (derived from durum wheat (*Tritcum turgidum* var. *durum* Desf.) or common wheat flour (*aestivum*) usually when supply of durum is limited or the price too high) mixed with water and mechanical energy input (mixing, extrusion, lamination) to produce a crumbly dough (~28–32% *w*/*w* water) either on an industrial or artisan scale producing a fresh pasta, which can then be dried. However, pasta has low amounts of dietary fibre, vitamins, essential amino acids and minerals [1] and during milling to make semolina there is some loss of these components. Pasta can be considered a good vehicle for including bioactive ingredients (proteins, phytochemicals, minerals, vitamins, etc.) as recognised by the World Health Organization and U.S. Food and Drug Administration because in some situations, up to 10–15% of non-traditional ingredients can be added without major loss of pasta quality depending on the ingredient and pasta processing technology employed [2,3]. However, the benefit of the added ingredient purported to provide, can be limited with such a low incorporation level. While designing foods with biologically active compounds, the resultant food often has technological deficiencies, undesirable appearance and sensorial properties making them less attractive to consumers or simply uneconomic to manufacture.

Increasingly many consumers in more wealthy economies are more interested in food that provides a benefit to prevent or reduce nutritional related diseases than was the case a decade or two ago. The main so called “lifestyle or civilisation” diseases have been associated with a combination of excessive caloric intake, poor nutrient balance and lack of sufficient exercise include obesity, overweight, elevated blood pressure, elevated blood cholesterol, cardiovascular disease (CVD), cancers, alimentary system disorders and type II diabetes mellitus (2-DM). These diseases afflict a large percentage of the population of westernized countries, with the trend continuing to worsen in developing nations and these diseases represent the main non-communicable cause of death [4]. The term functional food was developed first in Japan defined by the Food and Nutrition Board of the National Academy of Sciences as “any modified food or food ingredient that may provide a health benefit beyond that of the nutrients it contains” [5]. More common now is the consumer demand for food that not only provides additional health benefits but tastes good and has the texture and flavour close to the traditional food. Adding a functional ingredient will increase the cost of the food which, if a health benefit can be demonstrated, many consumers are likely to pay in discerning markets. 

Increasing demand by a growing number of health-conscious consumers for healthy foods has garnered interest from food manufacturers and a plethora of studies exist in the literature today [4,6]. Within the last decade or so, there has been a trend towards manufacturers trying to improve the nutritional value or create a presumed health benefit by supplementing semolina with various ingredients either as a partial replacement of the semolina or a complete replacement [3]. This approach is a powerful strategy for improving diet and wellbeing. Typical strategies employed to create a functional pasta are summarised in Table 1. 

Creating genetically modified wheat through transgenic or non-transgenic (TILLING, CRISPR-Cas9) or conventional breeding to modify starch, protein and lipid composition is an ambitious strategy gaining interest by breeders but is very much in its infancy. For example, a high amylose durum wheat was developed using TILLING showing that the GI of durum pasta, already low–medium, can be further reduced with the right starch biosynthetic enzyme mutation [7]. Beta-carotene has already been expressed in a genetically engineered rice cultivar, named Golden Rice to benefit people with vitamin A deficiency in developing countries [8]. Breeding durum wheat for enhanced functional or medicinal value is a new concept that has yet to take-off in commercial plant breeding programs and traditionally plant breeding of crop species has mostly focussed on yield, quality, biotic and abiotic tolerance but not food nutrition or health benefits. Research arising from the HEALTHGRAIN project between 2005–2010 described the genetic variation of key grain nutrient components and tools were developed for breeders for selecting cultivars with high levels of healthy compounds [9] but commercial application has been limited until recently. Some programs have begun work in this area and the reader is referred to the review by Yu and Tian [8]. Pasta made from pseudocereals (amaranth, buckwheat, quinoa) or blends with wheat flour are rich in micronutrients, phytonutrients, gluten free, with a more balanced amino acid profile than rice, wheat and maize, could be another route to have “ready-made” functional cereal products [10]. Ancient grains that can be readily made into flours can also provide some benefits although the evidence for their functional value is unclear [11]. Gluten free pasta can address the needs of the celiac person who must avoid gluten to prevent symptoms of this disease while gluten intolerant individuals, which are on the increase in society, will choose these products over gluten containing pasta. Simply adding various nutrients by substituting some of the semolina or adding them in the water used to make the dough at various levels (1–20%) of ingredient(s) is the most common strategy used and a focus of this review (Table 1 and Table 2). To improve the total protein of pasta (beyond the typical 10–15% range) or improve the amount of essential amino acids lacking in pasta (lysine, threonine, methionine), common sources used for pasta include legumes, cereal germ, dairy powders (egg white, casein), bovine serum powders, fish proteins and microbial fermentation products [6]. During the milling of grains many vitamins are lost which can be overcome by adding vitamin-rich tissues such as spinach, tomatoes, mushrooms, calf liver, sunflower seeds, chicken or fish meat [6]. Cereal bran is a good source of fibre, and vegetable oils, seafood and fish oil are excellent sources of polyunsaturated fatty acids. See Table 2 for some studies relating to enhancing limiting nutrients in pasta. 

Other avenues such as valorising cereal and non-cereal by-products is becoming more popular with a push to reduce food waste and create a circular food economy. Finally processing to modify the additive before its addition to the food, e.g., germination, fermentation, enzymic treatment, etc. is another approach to create a functional pasta. The aim of these strategies is to improve the nutritional and/or physiological functions from consuming such food on for example, gut health, immune system activity, mental status and/or to reduce the risk of specific pathologies for example cancer, cardiovascular disease, diabetes, and osteoporosis. 

In designing a functional pasta consideration must be given to the form the food is consumed. Pasta is traditionally eaten cooked after boiling in water for several minutes. In processing the extrusion or lamination process introduces forces, the drying process uses high temperature and variable humidity, so all these can affect the functionality of the included active ingredient(s). Even during digestion in the human system, there can be a loss in efficacy of the active ingredient (bioavailability) [12]. An added challenge is to ensure that the added ingredient should have a minimal impact on the pasta quality, palatability and consumer appeal. 

The present review aims to provide an overview of recently published scientific articles from the year 2000 to 2021 focussing on ingredients added to a durum wheat semolina –water formulation to make pasta with a functional benefit. However, where appropriate relevant pre-2000 references are included. Emphasis will be on more recent novel ingredients and evidence for demonstrated health benefits from animal or human trials rather than reports relying on the presumed benefit from only in vitro studies alone or assuming that the added ingredient’s known medicinal value transfers to the pasta consumed, which is not necessarily the case. While it might be a simple matter of adding an ingredient into the semolina–water mix at various percentages, the active ingredients functionality cannot be assured. 

Review of methods Recent reviews on the impacts on the technological quality of the pasta with added functional ingredients have been published in the last decade and the reader is referred to these to supplement this review [2,3,4,6,13,14,15,16,17,18,19]. Many of the studies mentioned in these reviews and other publications have been summarised in Table 2. Most of these have not shown evidence of the impact of the added ingredient on human health. A focus of this review was to present studies where some clinical evidence for the health benefit of the functional pasta (made only from durum wheat) is presented either in human or animal studies. For the other studies, a summary is provided in the form of a table (Table 2) where the author has decided to present relevant studies from the perspective of the raw ingredient(s) added to the pasta formulae, the likely active ingredient providing the purported health benefit, impact on the pasta quality and functional value with some prediction over the possible in vivo benefit if clinical studies were performed. Areas not covered in this review include the effect of functional pasta ingredients from ancient grains and pseudocereals [10]; gluten free pasta [20,21]; cereal and non-cereal by-products [17] and pasta made with agro-industrial by-products [13] to keep the scope of this review manageable. Database searches were supplemented by manual searches of the reference lists of included reports and previous reviews. Language was restricted to English only. The search strategy used was last updated on 5 October 2021 (Table 3). Studies using common wheat to prepare pasta were not included. 

**Table 1 foods-11-00123-t001:** Strategies to create pasta with added nutritional functionality.

Approach	Intention	Reference
**Genetically modify the composition of the grain**	Enhance a specific component of the grain	[7,8]
conventional breeding GMOs Gene technologies—TILLING, CRISPR-Cas9		
**Substitute semolina with various levels (1–20+%) of ingredient(s)** **with potential health value:**	Functional foods: Increase protein content and quality; increase fibre, AO, phytochemicals, etc.	[2,3,4,6,13,14,15,16,17,18,19]
Proteins, Fructo oligosaccharides, dietary fibre, Prebiotics, ω-3 fatty acids, minerals, vitamins Phytochemicals and probiotics Legumes (chickpeas, red lentils, yellow peas, faba beans, soy) Vegetables (pumpkin, zucchini, spinach, tomato) Herbs (oregano leaves, parsley leaves) Roots and tubers (cassava, sweet potato, beet, carrot) Others (gums, resistant starch, modified starch, β-glucan, psyllium seed husk)		
**Composite flours**	Reduce cost of pasta by replacing some/all durum semolina with common wheat flour and other flours	[15]
Common wheat and durum wheat Other cereal flours and durum wheat		
**Gluten free pasta**	Gluten free diet, celiac diet	[20,21]
**Processing to modify the additive before its addition to the food**	Enhance ingredient nutritional value, remove anti-nutritional factors	
Germination Fermentation Enzymic treatment Micronisation		[22] [6,23,24] [6] [25,26]
**Low value products/waste streams**	Valorisation of cereal and noncereal by-products	
Bran fractions Aleurone fractions Grape marc, fruit pomace Fish meal Whey Algae		[25,27,28] [29] [30] [31,32] [15,33] [34]
**Ancient grain/pseudocereals**	Valorisation of underutilised grains	[10,15,35]
einkorn emmer wheat kamut spelt buckwheat quinoa amaranth		

**Table 2 foods-11-00123-t002:** Examples of pasta made with a combination of semolina and non-traditional ingredients.

Ingredient Added	Active Ingredient	Substitution Range	Impact on Pasta	Predicted Health Benefits from Data Presented	Reference
**Barley fractions or β-glucan**					
Barley Balance^®^	β-glucan	0, 7.5, 15, 20	Provides AO, lowers IVSD, minimal impact on pasta making quality up to 7.5%	Lower GI, cholesterol reduction, SCFA production	[36]
Glucagel, Barley Balance^®^ (BB)	β-glucan	0, 2, 4, 6, 8, 10	BB only reduced IVSD. Some impact on pasta making quality after 4%	Lower GI, cholesterol reduction, SCFA production	[37]
Barley β-glucan fibre fraction	β-glucan	0, 2.5, 5, 7.5, 10	lowers IVSD but reduced pasta making quality above 2.5%	Lower GI, cholesterol reduction, SCFA production	[38]
Barley fractions	β-glucan	0, 5, 20, 40	Increased TDF, darker, acceptable sensory, lower total calories	unknown	[39]
Barley pearling fractions	β-glucan	50	Higher TDF, pasta darker with good cooking qualities	unknown	[40]
β-glucan enriched barley flour	β-glucan	40	Increased β-glucan to 5% in pasta; pasta quality comparable to control; higher AO and TPA; reduced IVSD	Could lower GI and enhance plasma oxidative defence	[41]
Barley hull air classification fractions	β-glucan	50% coarse Fr; 45% coarse Fr + 5% gluten; 95% coarse Fr + 5% gluten	Increased TDF and β-glucan; Higher AO and flavan-3-ols, TPA	Could enhance plasma oxidative defence	[42]
Soluble fibres: BB, psyllium, GR-inulin, HPX-inulin enriched pasta and doughs	Soluble fibres	15% individual fibre and dual combinations, 7.5% each	Pasta containing BB individually and in combination with psyllium showed an overall sensory acceptability comparable to control and in vivo glycaemic index reduction of 33–37%	Reduced pasta GI	[43]
Oat (1,3)(1,4) β-d-glucans	β-glucan	0, 5, 10, 15, 20	Oat β-glucan increased pasta water absorption, fat, TDF, and increased cooking loss >5%, minimal impact on appearance but sensory acceptable up to 15%. 10–15% Oat β-glucan and 5% additive of vital wheat gluten and xanthan gum yielded functional pasta containing 3.3–5.5 g β-glucans/100 g	Oat claim for lowering GI, lowering cholesterol	[44]
**Other dietary fibres components**					
Guar gum, CMC	Soluble fibres	CMC: 0, 0.25, 0.5, 0.75, 1.0, 1.5 GG: 0, 2.5, 5, 10, 15, 20	lowers IVSD with 20% GG but impacts pasta making quality; lowers IVSD with 1.5% CMC no impact on pasta making quality	Lower GI unknown level needed	[45]
Bran, pollard	Insoluble fibres	Bran: 0, 10, 20, 30 Pollard: 0, 10, 20, 30, 40, 50, 60	Up to 10% pollard can be tolerated minimal impact pasta quality with elevated AO, TDF. Bran had negative impacts pasta at all doses but with enhanced TDF, AO with no effect on IVSD	Higher TDF bowel health and transit	[27]
Commercial sources of pea fibre, Inulin, GG, locust bean gum, Xanthan gum, Bamboo fibre, HISol (B-glucan)	Non-starch polysaccharides, inulin	0, 2.5, 5, 7.5, 10	Increased cooking loss and reduced starch and protein and effects on texture varied with non-starch polysaccharides used and quantity with 5% the limit. Fresh pasta only used	Higher TDF bowel health and transit; lower GI likely, enhanced Ca absorption	[14]
Debranning fractions (DF) and micronized debranned kernels (MK)	AO, phenolics	DF 30%; MK 100%	Higher content of phenolic compounds with minimal effects on pasta sensory properties	Higher TDF bowel health and transit	[46]
Phenolic extract	Phenolics	Phenolic extract liquid replaces water used in pasta making	Dough was weakened, pasta was more brown and sensory scores impaired (more bitter and salty)	Poor strategy to enhance phenolics in pasta	[25]
Long-chain inulin (HPX), short-chain inulin (GR), Glucagel, psyllium and oat material added individually and in combinations	Inulin, β-glucan, dietary fibre	15	Addition of DF individually and in combination increased pasta optimum cooking time, cooking loss, water absorption and a deterioration in texture and colour values compared to non-DF enriched control. Oat bran flour with another DF gave the best pasta while psyllium fibre was the worst	Higher TDF	[47]
Wheat, rice, barley, oat brans	Insoluble fibres	0, 5, 10, 15, 20, 25	Decreasing sensory acceptability and colour and increase in cooking loss with increasing dose, least impact with oat bran	unknown	[28]
Dephytinized rice, rye, wheat, oat	Insoluble fibres	20	A 1.7–2.9% increase in pasta TDF. Increased TPA andF AO and Ca, P, K, Mg, Zn with significantly reduced phytic acid content	Higher TDF bowel health and transit	[48]
Whole wheat durum pasta	Wholegrain components	Whole wheat vs. regular pasta	Whole wheat dough is weaker, pasta is reddish-brown with higher cooking loss, lower firmness in cooked product and reduced mechanical strength of dried compared to regular pasta	Potentially multiple benefits, likely lower GI	[49]
Micronized wheat bran with CMC, XG, locus bean gum	Insoluble fibres and gums	11.5	Egg tagliatelle pasta with added XG > 0.8% improved textural properties and CMC >0.6% to enhance yellowness was found to produce a healthier pasta product with higher content of fibre, minerals and vitamins and suitable quality	Higher TDF and potential health benefits from this	[26]
High fibre oat pasta	Soluble and insoluble fibre	10, 20	Oat fibre increased pasta TDF ~8% but increased water absorption and cooking loss, decreased brightness and firmness and impacts reduced using fine (volume mean diameter, μm 50.5) vs. medium (141) and coarse (249) oat powder	Oat claim for lowering GI, lowering cholesterol	[50]
**Fish products and algae**					
Spirulina microalgae enriched pasta	water-soluble pigments and phycocyanin and phenolic compounds	3	The technological properties of pasta were affected, but overall acceptability index (85.13%) not influenced by microspheres. Microencapsulated spirulina protects the microalgae’s antioxidant potential	Benefits from AO	[34]
Pastas with added concentrates of flesh and skin from aquaculture seabass	Source of polyunsaturated fatty acids and minerals	concentrate fish flesh powder 10, concentrate fish skin powder, 20	Increased Ω-3 fatty acids in pastas with fish concentrates, decrease in the Ω6/Ω3 ratio that greatly exceeds current nutritional guidelines. All pastas showed a low valuation in negative attributes such as oil, or rancidity flavours. Main differences detected were colour, fishy flavour, odour, and texture (chewiness)	Possible improved cardiovascular health markers	[51]
Salmon fish (*Oncorhynchus tschawytscha*) powder (SFP) supplemented pasta	Antioxidants and other carotenoids	5, 10, 15, 20	SFP addition to pasta increased the release of phenolic compounds and AO activity from pasta during digestion to achieve higher levels than control pasta and also reduced the in vitro starch digestibility	Lowers GI	[31]
Pasta formulation was substituted with shrimp meat	Omega-3 polyunsaturated fatty acids	10, 20, 30	shrimp meat (*P. monodon*) can be added up to 20% without drastically affecting the sensory attributes of pasta with enhanced nutritional quality (protein, fat and ash content)	unknown	[32]
**Herbs**					
Dried amaranth leaves and amaranth seed flour pasta	Peptides derived from protein, source AO, phenolic acids, flavonoids, carotenoids	amaranth seed flour, 21.25–50.97% and dried amaranth leaves, 0–5.61%.	Pasta with amaranth seed flour and dried amaranth leaves exhibited significantly higher content of protein, crude fibre, minerals with higher AO but panellists preferred pastas with low percentage levels of amaranth seed flour	Benefits from AO anti-hypertensive, anti-oxidant, antithrombotic, anti-proliferative, and anti-inflammatory activities	[35]
Wild edible plants, *Pereskia aculeata* Miller or American gooseberry dried leaf flour enriched pasta	Source of protein and lysine, soluble fibre, minerals, vitamins	0, 10, 20	Improved pasta dietary fibre, calcium, iron compared to the control pasta. Enriched pasta presented a greenish fibrous appearance. Sensory evaluations indicated that pasta enriched with 10% did not affect consumer acceptance	constipation, obesity (high satiety due the dietary fibre content)	[52]
**Inulin addition**					
Inulin enriched pasta	Inulin	0, 2.5, 5, 7.5, 10, 20	The higher molecular weight inulin had minimal impact on pasta quality and sensory properties until 20% while lower MW inulin had more negative impacts on pasta firmness, cooking loss, and sensory acceptability. IVSD was reduced in pasta with inulin higher MW inulin up to 5% but was increased with 20% inulin. Inulin enhanced the gluten structure in pasta with higher starch crystallinity	Lower GI	[53]
Fresh pasta with inulin (FRUTAFIT HD)	Inulin	0, 2.5, 5, 7.5, 10	Inulin was shown to influence the swelling index and firmness, but not the adhesiveness and elasticity of pasta products and lowered IVSD	Lower GI	[54]
**Legume addition**					
Chickpea flour	phytic acid, sterols, tannins, carotenoids, as isoflavones	5–20	Increased protein content; sensory properties (colour, flavour and overall acceptability) improved up to 10%; >30% led to lasagne processing handling and cooking characteristics deterioration and soft mushy pasta	Higher quality protein with good balance of amino acids	[55]
Desi chickpea ‘besan’ flour	phytic acid, sterols, tannins, carotenoids, as isoflavones	0, 10, 15, 20, 25, 30	Up to 15% chickpea can be tolerated in spaghetti with acceptable pasta making quality	Higher lysine and protein content	[56]
Legume pasta (mung, soya, red spit lentil, chickpea)	Soluble and insoluble fibres	10	No negative impact on technological quality or IVSD	Higher TDF and potential health benefits from this	[57]
Black chickpea flour and fermented black chickpea dough pasta	phytic acid, sterols, tannins, carotenoids, as isoflavones	5.6 (Black chickpea flour), 15 (black chickpea dough)	Fermentation enabled release of 20% of bound phenolic compounds in the dough, higher resistant starch and total free amino acids while antinutritional factors significantly decreased. Fortified pasta had higher in vitro protein digestibility (up to 38%) and higher AO levels. Fermentation reduced antinutritional elements in the black chickpea flour. Sensory acceptance while different to control described a peculiar but appreciated profile of the fortified samples, especially for the pasta including fermented black chickpea dough.	unknown	[58]
Lentil flour and CMC	proteins, dietary fibres, oligosaccharides, starch, polyphenols, fatty acids, and antioxidants and prebiotics	40 (lentil) 2 (CMC)	Lentil fortified spaghetti increased essential amino acids but caused a decrease in pasta quality (e.g., higher cooking loss, lower breaking energy) that was improved by adding CMC	unknown	[59]
Mexican common bean flour	proteins, vitamins, complex carbohydrates and minerals	0, 15, 30	The cooking time and water absorption decreased and cooking loss increased to unacceptable levels, firmness decreased and pasta was darker as a function of the bean flour percentage. Protein increased. Increases of furosine and marginal increases in phenolic contents in pasta	Benefits from TPA	[23]
Faba bean pasta	Essential amino acids	0, 30, 70, 100	Faba enriched pasta weakened the protein network that could be responsible for the increase in the in vitro protein digestion but led to high cooking loss and reduced resilience in cooked product. Very high temperature drying strengthened the protein structure of pasta, resulting in increased integrity and better resilience of pasta without altering their in vitro protein digestibility. Appreciation of legume pasta containing 80% or 100% was similar to that of commercial whole wheat pasta	unknown	[60]
Pasta with added chickpea flour	Fibre, proteins	20, 40	Protein, ash, lipid, and dietary fibre and RS content increased by adding chickpea flour to the pasta. The starch hydrolysis index decreased as chickpea flour in the pasta increased, with a lower predicted glycaemic index than durum wheat-control pasta.	Lower GI	[61]
Yellow pea pasta	alkaloids, flavonoids, glycosides, isoflavones, phenols, phytosterols, phytic acid, protease inhibitors, saponins, tannins	0, 10, 20, 30	20% yellow pea flour had favourable sensory attributes, protein content, good texture, yellowness values, reduction in the glucose release and increased protein digestibility. Dough was weaker while product appearance similar to control	Lower GI	[62]
Pasta with split pea and faba bean	Fibre, protein, vitamins and minerals	35	Increased cooking loss, lower pasta breaking energy, altered sensory properties (higher hardness and fracturability). High drying temperature improved slightly but pasta redness increased to undesirable levels with very high T drying	Higher TDF and potential health benefits from this	[63]
Pasta with added germinated pigeon pea (*Cajanus cajan*)	low fat, fibre, proteins and starch, balanced of minerals	0, 5, 8, 10	Germination of pigeon pea reduced antinutritional components and increased vitamin B2, E and C. Good acceptability, higher protein, total available sugars, dietary fibre, micronutrients, and vitamins than pasta made from 100% semolina but impacts on pasta making quality (shorter cooking time, higher water absorption and cooking loss)	Vitamins, fibre, better protein balance	[22]
Corn gluten meal enriched pasta	High protein source	0, 5, 10	Corn gluten meal increased pasta protein content, had a similar cooked weight and cooking loss but was less firm with inferior colour compared with the control. The overall flavour quality score of the spaghetti decreased	Unknown	[64]
**Lupin addition**					
Lupin flour to replace semolina	High protein and fibre source	0, 10, 20, 30, 40, 50	Minimal impacts on pasta cooking loss and dry pasta colour and no difference in sensory acceptability up to 20% but α-galactosides and antinutritional factors like phytic acid, saponins, lectins and protease inhibitors reduce protein digestibility	unknown	[65]
Lupin protein isolate	Proteins, AO, TDF	0, 5, 17, 30	Lupin protein isolate increased protein up to 129%, reduced pasta cooking time, water absorption and cooked firmness while stickiness and cooking loss were increased. Lupin protein isolate made the dried pasta more red and yellow and decreased brightness. The percentage of starch digested under in vitro conditions was reduced using 17% lupin protein isolate	Reduced GI	[66]
α-galactosides free lupin flour	High protein and fibre source	0, 50, 80, 100	α-galactosides free lupin flour can improve pasta nutritional value without flatulent causing oligosaccharides	unknown	[22]
**Protein addition**					
Lupin protein isolate	High protein and fibre source	0, 5, 17, 30	Lupin protein isolate increased protein, reduced cooking time, water absorption and firmness but stickiness and cooking loss increased making dried pasta duller	unknown	[66]
Durum bran protein concentrate	High in phytosterols, protein and EAA	0, 1 5, 10, 20	Pasta quality acceptable up to 10% and enriched in EAA	Benefits from better protein quality	[67,68,69]
Whey enriched pasta	High in protein and EAA	0, 20	Whey addition increased protein content, and pasta water uptake with minimal impact on sensory quality	unknown	[33]
Beef lung powder enriched pasta	High in protein and EAA, Fe	0, 10, 15, 20	Pasta had higher cooking loss, cooked pasta was firmer and much darker than control with reduced IVSD, higher Fe and protein content	Lowers GI	[70]
Mustard protein isolate enriched pasta	High in protein and EAA	0, 2.5, 5, 10	Increased pasta protein while cooking loss, cooked weight and stickiness decreased and firmness increased while pasta is duller and more red	Unknown	[71]
*Phaseolus vulgaris* protein hydrolysate	angiotensin I-converting enzyme inhibitory activity (ACE) and AO	0, 5, 10	Pasta with bean had higher protein content with good sensory acceptability up to 10% with ACE and AO activity	BP regulation	[72]
**Resistant starch**					
Hi Maize™ RSII and Novelose 330™ (RSIII) enriched pasta	Resistant starch	RSII: 0, 10, 20, 50 RSIII: 0, 10, 20	Minimal impact on pasta quality using these ingredients up to 20% while increasing RS content of pasta, stable after cooking. Both RS reduced IVSD	Lower GI gut health benefits from RS	[73]
Hi Maize260™, Hi Maize1043™, RSII and Fibersym70™ (RSIV) enriched pasta	Resistant starch	0, 10, 20	RS addition had minimal impact on pasta quality and acceptability while reducing the IVSD	Lower GI gut health benefits from RS	[73,74]
Unripe banana fibre	Starch from unripe banana flour	0, 5, 10, 15, 20	Increased pasta RS, decreased gluten, was darker, higher cooking loss and firmness lower while sensory analysis found banana starch improved acceptability up to 15% but this analysis was limited	Unknown	[75]
Pastas with elderberry juice Concentrate (EJC) and Hi-maize starch or apple pectin	phenolic acids, anthocyanins, flavanols, carotenoids, vitamins and minerals, soluble DF	10 g Hi-maize starch, pectin or combination, and diluted elderberry juice concentrate (50 mL per 50 g flour)	Adding EJC to fettuccine pastas reduced the firmness, wettability and volume expansion of the fresh pastas, but Increased protein, total DF content, total antioxidant activity and total extracted TPA content	AO and TDF mostly from insoluble fibre	[76]
**Vegetables**					
Stems of *Opuntia ficus-indica* (cladodes), dried and ground and extracted (*Opuntia cladode* extract )	Rich in soluble fibre (arabinose, galactose, rhamnose, xylose and galacturonic acid)	0, 10, 20, 30 mL substituting the added water used to prepare pasta	Comparable quality and sensory acceptability using up to 10–20% *Opuntia cladode* extract. IVSD decreased with increasing level of *Opuntia cladode* extract and cholesterol bioaccessibility decreased which could reduce blood cholesterol	Blood cholesterol- and glucose-lowering capabilities	[77]
Carrot leaf meal and Oregano leaf meal	alpha-linolenic acid, omega-3 fatty acids	0, 5, 10 of each and combinations	Increased AO, and omega-3 fatty acid content from as little as 5% but pasta with higher cooking loss, shorter optimum cooking time, reduced weight increase but all formulations were acceptable by sensory the best being 10% oregano and carrot leaf meal	Unknown	[78]
Soy okra soybean by-product	protein, lipid, dietary fibre isoflavones, phytosterols, coumestans, lignans, phytates, and saponins	0, 10, 20, 30, 40	Increasing soy okra flour reduced pasta optimum cooking time, increased cooking loss and altered taste, texture and colour tolerating only 10%. However, AO and total phenolic contents increased and predicted GI (IVSD) decreased	Lower GI, TPA presumed benefits	[79]
Mushroom powder (white button, shiitake and porcini)	proteins, acidic polysaccharides, dietary fibre and antioxidants	0, 5, 10, 15	mushroom powder increased pasta cooking loss and cooked firmness The addition of shiitake mushroom powder resulted in pasta with the highest firmness and tensile strength	unknown	[80]
Tomato peel	Antioxidants, carotenoids, DF	0, 10, 15	Detrimental effect on pasta such as colour, break resistance, high firmness, reduced cooking loss, inferior sensory taste and overall quality at 10% and higher. However, by adding CMC or gums could negate some of these effects on sensory. Nutritionally tomato peel enhanced b-carotene, lycopene and TDF	to scavenge reactive oxygen species and protect against degenerative diseases like cancer and cardiovascular diseases	[20]
Onion powder	Flavonoids, Quercetin, Proteins, saponins and phenolic components	0, 5, 10, 15	Onion powder up to 10% does not affects sensory characteristics and provides 2.2 mg/100 g of quercetin	Unknown	[81]

**Table 3 foods-11-00123-t003:** Database search history showing database, search term and number of hits.

Web of Science (2000–2021)									
Pasta and human health	Glycaemic index and pasta	Cardiovascular disease and durum pasta	Diabetes and durum pasta	Obesity and durum pasta	Insulin and durum pasta	Dietary fibre and durum pasta	Cholesterol and durum pasta		
27	19	8	8	5	8	53	4		
PubMed (2000–2021)									
Pasta and health	CVD and durum wheat pasta	diabetes and durum wheat pasta	obesity and durum wheat pasta	weight gain and durum wheat pasta	cancer and durum wheat pasta	insulin and durum wheat pasta	cholesterol and durum wheat pasta	dietary fibre and durum wheat pasta	dietary fibre and durum wheat pasta and health
707	6	5	5	3	3	15	8	62	20
Cochrane Registry									
durum pasta and obesity	durum pasta and CVD	durum pasta and weight gain	durum pasta and cancer	durum pasta and insulin	durum pasta and dietary fibre and health	durum pasta and cholesterol			
0	1	2	1	6	0	1			

## 2. Results and Discussion

### 2.1. Health Based Evidence for Functional Pasta

Different approaches can be used to enhance the nutritional and potentially health promoting properties of pasta. Consuming pasta simply as a wholegrain or wholemeal product which has been available commercially for many years is probably the simplest way. Wholemeal pasta is when bran has been added back to the semolina, while wholegrain pasta refers to “the intact, cracked, ground or flaked caryopsis, whose anatomical parts, endosperm, bran and germ are found in the same quantity as present in the intact original grain” [82]. Since the bran and germ contain many biologically active compounds such as vitamins, minerals, essential fatty acids, amino acids and many phytochemicals, they have been linked to reducing the risk of many lifestyle diseases [83]. However, consumer preference is for pasta made from refined semolina or flour due to better taste, appearance and texture despite fewer health benefits compared to wholemeal/wholegrain counterparts. Other approaches to enhance the nutritional properties of pasta include the addition of specific ingredients or combination of ingredients to provide specific functionalities based on knowledge about their function in isolation or from research studies in other foods. To enhance the consumption of pasta with added health benefits researchers, industry and relevant agencies need to overcome some of the barriers to their uptake such as improving the sensory quality, processing issues (cooking time), availability and media mixed messages. 

Studies where health promoting ingredients have been added to pasta and evidence of a health affect using animal, human clinical studies or in vivo measurements are discussed. These are divided into the major disease risk categories. 

#### 2.1.1. Hypoglycaemic Effects

Lowering the absorption of carbohydrate into the blood stream from the intestine has been shown to reduce the risk of developing metabolic disease and type II diabetes mellitus (2-DM) while lowering insulin demand caused by eating slowly absorbed carbohydrates less likely to induce insulin resistance in healthy people [84]. It has been demonstrated in vitro and in vivo that durum wheat pasta made from a high amylose durum wheat (with elevated resistant starch), at least above ~50%, reduces the postprandial glycaemic response (PPGR) compared to regular durum wheat pasta with amylose 25–30% [7]. Food structure plays an important role in determining a foods glycaemic response. Both the compact structure of pasta and the presence of the gluten network which surrounds the starch granules together interferes with α-amylase breakdown of the starch is thought to be the mechanism for this effect [85,86,87]. Similarly in noodles [88] fed 12 healthy subjects noodles with amylose contents ranging from 15–45% obtained by blending high amylose wheat flour and showed a reduction in the PPGR in the 45% amylose noodles compared to the low amylose, 15% noodles but no difference to the 19.6% amylose noodles. This is supported by earlier studies substituting semolina for high amylose (>75%) maize flour in pasta with significantly lower PPGR and postprandial insulin levels [89]. 

Other ingredients added to pasta have also shown a reduction in the glycaemic response and some studies are discussed. Lupin (*Lupinus albus*) flour contains a protein called ϒ-conglutin, shown to decrease glycaemia in humans and an extract enriched in this protein was added to pasta (125 mg of pure protein in 100 g of pasta) which was fed to hyperglycaemic rats as uncooked food for three weeks. The protein enriched pasta with ϒ-conglutin led to a decrease in food intake, and a reduction in glycaemia [90]. Authors noted that their results could have been affected by the lower carbohydrate content in the lupin meal with respect to the control and that the pasta was not cooked before feeding as the stability of the ϒ-conglutin protein could have been affected. Goñi and Valentín-Gamazo (2003) [91] fed 12 healthy subjects test meals of durum spaghetti and spaghetti containing 25% chickpea flour and the latter had significantly lower GI than the regular spaghetti as well as increasing the mineral, fat and indigestible content of the pasta. Authors suggested this was due to the presence in the chickpea flour of non-starch polysaccharides resistant to enzymic digestion. 

Soluble dietary fibres such as β-glucan, guar gum, psyllium and alginate can reduce elevations in postprandial glucose [92,93] because of their viscosity properties which adjusts the rate of gastric emptying. The insulin response of 11 healthy males fed a high fibre pasta made from 40% barley flour high in β-glucan was compared to regular wheat flour pasta. Carbohydrate was more slowly absorbed from the high fibre pasta with a reduced insulin response [94]. Some of these components are already present in certain foods (β-glucan rich sources are oats and barley; Plantago ovata plant for psyllium) and efforts to isolate these fibre components for use as supplemental dietary fibre in functional food design is attractive. To be certain of the effectiveness of the active ingredient, food processing and the form of food consumption (cooked, steamed, etc.) may modify the food structure, ingredient stability and fibre viscosity and potentially impact any proposed health claims. Thus, food manufacturing process may or may not preserve the beneficial properties of the added ingredient and should be considered. 

Pasta made from debranned durum wheat flour, enriched in polyphenols and with added barley β-glucan and *Bacillus coagulans* GBI-30, 6086 (probiotic) had good cooking quality with high content of bound ferulic acid compared to control pasta. The probiotic strain remained viable during the pasta-making and cooking processes. However, the PPGR measured in healthy volunteers was no different to control pasta [95]. Frost et al. (2003) [96] included soluble fibre psyllium into pasta to see the impact of a viscous fibre fed to 10 subjects. While there was no effect on gastric emptying or the incremental area under the curve for glucagon-like peptide 1 compared with the control pasta, the added polyunsaturated fat (30 g) and sodium propionate (3 g) in the pasta recipe did alter these parameters which could reduce the risk of diabetes and improve coronary risk factor profiles. Authors suggested the combined high-fat meal with psyllium-enriched pasta may affect the intestinal milieu, affecting carbohydrate digestion and glucose uptake from the small intestine with slower rates of gastric emptying [96]. The addition of fat to a food can reduce glucose response to carbohydrate. 

Evidence for efficacy of soluble fibres on PPGR in other foods is extensive [97] but there are issues with their application particularly with regards to sensory acceptance, due to the requirement for relatively large quantities necessary to confer the intended health benefit. To maximize the bioavailability and physiological effects of soluble DF in relation to PPGR, functional food design and assessing processing effects is needed. For example, during extrusion there are forces and heat developed that can reduce the soluble fibre molecular weight, reducing viscosity and effectiveness on PPGR [98]. The most effective soluble fibre from clinical studies in attenuating the PPGR when consumed with a high carbohydrate food like pasta seems to be β-glucan provided it undergoes minimal processing [99]. This efficacy can be diminished with food processing for example Bourdon et al. (1999) [94] found no effect on PPGR when β-glucan was added to pasta because the food structure was not altered by the food processing. More research is needed to develop food manufacturing procedures that minimise disruptions to pasta structure and the resulting viscosity. While a positive effect on PPGR in clinical studies is desirable, longer clinical trials are needed to establish a link between attenuation of blood glycaemia and a reduction in incidence of lifestyle diseases related to PPGR. 

Taha and Wasif (1996) [100] fed diabetic rats a diet consisting of semolina only pasta, or wholemeal pasta or wholemeal pasta supplemented with 12% soy flour and 3% methionine for 28 days. They showed that the latter pasta diet lowered total glycerides and cholesterol, and within 10 d, it lowered the PPGR compared to rats fed only semolina or wholemeal pasta, which was maintained at a lower level over the study period. Using wholegrain pasta as a control, pasta containing barley β-glucans and *Bacillus coagulans* BC30, 6086 were fed to healthy overweight or obese volunteers (n = 41) for a 12-week intervention study. The study found that a daily serving of symbiotic whole-grain pasta reduced glycaemia (plasma high-sensitivity C-reactive protein) and plasma LDL/HDL cholesterol ratio [101]. 

Recently, a review of the GI of 74 pasta products consisting of refined and wholewheat pasta made from durum semolina or white wheat flour, together with pasta made with added egg or legumes or vegetable or algae or other ingredients were described. This database of pasta GI studies (minimum 10 subjects) show a large variability with GI ranging from 18 to 93. Most pasta products had low to medium GI with the median value of 52.5, which is low GI < 55 by definition [102]. The variability within each group reflects the different processing methods for manufacturing, and different subject groups and laboratories conducting the GI test, but, overall, the review concludes that pasta is generally a low GI food. Details on the influence of pasta processing on starch digestion is discussed elsewhere [103]. 

#### 2.1.2. Hypocholesterolemic Properties and Beneficial Effects on Cardiovascular Disease (CVD)

Attempts to reduce the risk of CVD with diet are varied and aim to prevent the move towards use of drugs which impart their own risks. Recent guidelines recommend consumption of functional foods with evidence from epidemiological studies indicating adequate consumption of whole-wheat or wholegrain foods is associated with reduced CVD risk [104,105]. Favari et al. (2020) [106] fed 41 subjects daily for 12 weeks a whole-wheat pasta (control) and a new innovative whole-wheat pasta enriched in barley β-glucans (2.3 g/100 g) and supplemented with spores of *Bacillus coagulans* GBI-30, 6086 (10^8^–10^9^ CFU/100 g). They showed improvement in serum cholesterol efflux capacity in overweight/obese participants, indicating the potential of a functional food to improve athero-protective high-density lipoprotein cholesterol function. Patients with hypercholesterolemia fed a soy-germ-enriched pasta containing isoflavone aglycons displayed improved serum lipid markers of cardiovascular risk [107]. A similar study in patients with T2D [108] showed the same soy-germ-enriched pasta significantly reduced blood pressure, and oxidative stress thought to be due to the high antioxidant capacity of the isoflavones in soy protein [109]. 

Use of non-live bacterial cells (paraprobiotics defined as inactivated microbial cells or cell fractions) as alternative to probiotics decreases risks in certain individuals and avoids need to use dairy foods as a delivery vehicle. Since pasta is processed and consumed after heat treatments, use of paraprobiotics has an advantage over probiotics. Almada et al. (2021) [110] investigated the effects of consumption of wheat-durum pasta with added *Bifidobacterium animalis* inactivated by gamma-irradiation on the health and gut microbiota of rats. Durum wheat pasta with added *B. animalis* was prepared, cooked and dried and the ground material fed to rats for 15 days. This pasta was found to reduce the serum glucose and total cholesterol levels in healthy rats compared to a standard control (non-pasta) and changed the gut microbiota. Pasta can be an effective vehicle to deliver this paraprobiotic. 

A common pre-biotic, inulin (a fructan carbohydrate) has been shown to reduce serum triglycerides that might help reduce the development of the metabolic syndrome. Inulin (Raftline HP = Gel) was incorporated into pasta (11%) and together with regular 100% semolina control pasta fed to 22 healthy males in two 5 week feeding periods in a crossover design. Inulin enriched pasta improved lipid (reduced triglycerides and increased HDL-cholesterol) and glucose metabolism (lower fasting glucose and haemoglobin A1c) and delayed gastric emptying. Delayed gastric emptying could be caused by colonic fermentation of the inulin leading to short chain fatty acid production inhibiting gastric emptying [24]. Slowing the gastric emptying can also decrease glucose absorption of foods, reducing PPGR. Indeed, improved metabolic control in the group treated with inulin-enriched pasta was observed. This level of inulin addition (11%) from other studies has been shown to have a minimal impact on traditional pasta quality measures depending on the degree of polymerisation of the inulin used [53]. No side effects on the gastrointestinal tract were found in the study [24]. 

*Opuntia ficus-indica* (prickly pear) is an important source of vitamins C, B1, B2, A, and E and minerals such as potassium, calcium, magnesium, and phosphorus. Durum wheat pasta was supplemented with 3% *Opuntia* and fed to 49 people with metabolic syndrome for 4 weeks. Improved atherogenic benefits were obtained such as reduced waist circumference, plasma glucose and triglycerides indicating beneficial effects of this extract [111]. 

In a randomised controlled trial consisting of meals of regular pasta (control) or pasta with 40% sprouted chickpea flour fed to 22 participants, a higher AO content and brachial artery flow-mediated dilation was achieved eating the functional pasta indicating potential benefits to cardiovascular health [112]. 

#### 2.1.3. Antihypertensive Effects

In a recent study Valdez-Meza et al. (2019) [113] prepared pasta at different protein contents with amaranth protein concentrate and an amaranth hydrolysate to evaluate antihypertensive properties in rats compared to regular pasta. The antihypertensive amaranth activity of the hydrolysate was maintained after incorporation in the pasta and after pasta ingestion, reducing blood pressure in the rats, confirming bioavailability. These additives reduced the sensory desirability of the pasta as assessed by 30 untrained panellists compared to regular pasta. Hydrolysis of amaranth proteins with microbial alcalase can release ACE-1 inhibitory peptides that can reduce the activity of angiotensin-1-converting enzyme which is involved in the pathogenesis of hypertension. The presence of these proteins in pasta was evaluated as a vehicle for consumption of these proteins in a food matrix. Pasta was supplemented with an alcalase-treated amaranth protein concentrate and compared to regular pasta. This ingredient negatively impacted the overall acceptability but antihypertensive measures in rats indicated reduced blood pressure [113]. 

#### 2.1.4. Oxidative Stress and Aging Effects

Oxidative stress is a condition where there is an imbalance between the generation of free radicals, such as reactive oxygen/nitrogen species, and the antioxidant defences (endogenous antioxidants glutathione, catalase and superoxide dismutase). Lack of dietary intake of foods rich in antioxidants, such as polyphenols, can play a role in the development and progression of many chronic diseases, such as CVD [114,115,116] and diabetes. While some information exists on the level of AO in pasta enriched in various ingredients [4,117] impacts on the AO status in vivo is lacking for most functional pasta studies. Epidemiological studies have shown an inverse association between the consumption of polyphenolic-rich foods and the risk of chronic diseases associated with oxidative stress [118]. Khan et al., 2014 [119] fed cooked pasta containing red wholegrain sorghum flour (30% *w*/*w*) to 20 healthy subjects and found elevated levels of polyphenols, antioxidant capacity and superoxide dismutase activity in their blood compared to 100% semolina pasta control thus improving the antioxidant status. This level of incorporation was found to be acceptable to consumers [119]. 

Laus et al., 2016 [120] fed 7 healthy subjects pasta bran enriched in lipophilic antioxidants or bran enriched in phenolics compared to non-supplemented pasta control. These pastas were similar in sensory score to control pasta. Lipophilic pasta improved the antioxidant status of the serum similar to a wheat AO rich commercial dietary supplement called Lisosan G while the phenolic antioxidants enriched pasta effected serum AO status. There were no differences in the AO status of the pasta extracts by in vitro assay. 

Pasta enriched with tartary buckwheat (*Fagopyrum tataricum* Gaertn.) sprouts (30%) was characterised by a high quercetin content and antioxidant activity. When fed to rats for six weeks, the rats exhibited a significant decrease in DNA damage (38%) and more efficient DNA repair (84%) compared to rats fed with commercial pasta [121,122]. Pasta enriched with 6% β-glucan can lower oxidative stress in people based on a longitudinal study that lasted 30 days [123]. 

Healthy diets have been linked to delaying the onset of aging disabilities and pathologies. Cactus pear extract was added to pasta (3% *w*/*w*) and fed to healthy human subjects for 30 days which led to decreased glycaemic and anti-inflammatory responses with putative effect on the aging process and related metabolic disorders [124].

#### 2.1.5. Other Effects

Weight gain and obesity are critical societal issues facing many communities world-wide and the push for foods that are less energy dense and promote satiety is strong. Pulse flours are higher in protein than cereals and contain slowly digestible and resistant starch. They also provide a better amino acid balance with higher levels of cereal deficient lysine and threonine. Up to 35% faba bean flour has been incorporated into pasta but can reduce pasta quality thought to be related to structural impacts [63]. These and possibly the presence of α-amylase inhibitors may explain the slowing of starch digestion and a lower postprandial glycaemic response. Faba bean flour and protein concentrate were added to pasta (25% dwb) and fed to 15 human subjects and compared to a durum semolina pasta. Pasta with faba bean added had reduced postprandial blood glucose response and improved satiety with acceptable sensory liking for the faba bean flour pasta [125]. Greffeuille et al. (2015) [126] over a two and a half month period fed 15 healthy subjects cooked durum wheat pasta dried at a low temperature (control), and pasta enriched with 35% faba bean dried at either a low or very high temperature and the GI was determined and visual analogue scale (degree of fullness). Inclusion of 35% faba bean flour in pasta increased resistant starch content but had no effect on starch digestion extent in vitro or the in vivo GI, despite disruption to the pasta structure. Using a high-temperature drying cycle during pasta manufacture but with no impact on pasta GI did improve its global digestive comfort and led to a decrease in appetite after eating. 

A recent review and meta-analysis of randomised controlled trials of pasta consumption in adults showed a significant reduction in body weight gain and body mass index compared with higher GI dietary patterns, dispelling the myth that a carbohydrate staple food such as pasta is a cause of the obesity epidemic [127].

In a study by Costabile et al. (2018) [128] a randomized, controlled, crossover trial (14 subjects) consumed whole grain (with 13% higher TDF) instead of refined wheat pasta and this improved appetite control but did not influence acute energy balance. After the wholemeal pasta, the desire to eat and the sensation of hunger were lower (−16%, *p* = 0.04 and −23%, *p* = 0.004, respectively) and satiety was higher (+13%; *p* = 0.08) compared with the control pasta. After consumption of wholemeal pasta, the blood glucose and triglyceride levels increased compared to control pasta. Insulin response at 30 min (*p* < 0.05) and ghrelin at 60 min (*p* = 0.03) were lower and PYY (anorexigenic gastrointestinal hormone Peptide YY) levels higher (AUC = +44%, *p* = 0.001) in subjects that ate the wholemeal compared to the refined wheat pasta. 

Fibres can be used to reduce digestion and absorption in the human small intestine and thus reduce the daily caloric intake [16]. Typical DF used in pasta are legume fibre, wheat bran insoluble fibre, inulin, psyllium fibre, olive powder, psyllium seed husk, oat β-glucans, *Lentinus edodes* β-glucans, resistant starch, common bean flour, and some non-starch polysaccharides such as locust bean gum, xanthan gum, guar gum, and pectin (Table 2). 

### 2.2. Pasta with Added Functional Ingredients with No Direct Evidence of Health Benefits

There is a plethora of reports on adding ingredients into a pasta formula without any evidence of health effects from animal or human studies [2,3,4,6,13,14,15,16,17,18,19]. Rather than repeat the approach taken in these reviews, a summary of studies where durum only semolina has been substituted in part with an ingredient are listed in a table grouped into arbitrary categories: barley components, dietary fibre, fish products, herbs, inulin, legumes, oat, proteins, resistant starch, soy and vegetable (Table 2). For each of the listed studies (n = 60), information on the ingredient added to the pasta, the amounts, the likely active ingredient(s), the reported impact on the pasta quality and the authors predicted health benefits from the data are presented (Table 2).

The majority of the studies reviewed in the previous section focus mostly on evidence for reducing the PPGR in humans, a few on CVD risk reduction and blood pressure brought about by a range of pasta supplemented ingredients (β-glucan, soluble fibres, high amylose flours, inulin) and other flours from other crop species (chickpeas, sorghum, pseudocereals, faba bean, lupin). No other health conditions seemed to have been looked at with functional pasta to date that meet the search criteria and inclusion restrictions. For studies where barley fractions or commercial β-glucan has been included in the pasta recipe, the in vitro studies show a reduction in starch digestion extent [36,37,38,41] that compares with the in vivo data from clinical studies [43,92,93,94]. Amounts less than 20% are generally effective in reducing in vitro starch digestion however there are impacts on the pasta quality especially above 10% if using β-glucan extract or commercial sources (e.g., Barley Balance^®^) but this depends on the β-glucan and if used in combination with other ingredients like vital wheat gluten or gums that can overcome some limitations of β-glucan [44]. Although Peressini et al. (2020) [43] confirmed differences in sensory attributes between Barley Balance^®^ enriched (15%) pasta samples and control pasta, these differences were not judged detrimental for the overall quality.

Inulin (a prebiotic) addition to pasta has been shown in laboratory studies to lower starch digestion up to ~5% inclusion [53] and at higher levels > 7.5% [54] backed up by clinical studies showing 11% inulin pasta slows the gastric emptying causing a decrease in PPGR [24]. Clinical studies adding flours from chickpea [91,129], faba bean [125,126], lupin [92] and red lentil flour [130] to pasta show reductions in GI. Laboratory studies with these flour additions to pasta support reduction in GI [61] while similar in vitro studies in faba bean are lacking and only one study looked at pasta fortified with lupin protein isolate (17%) showing a reduction in in vitro starch digestion [66]. Soy has also been shown in clinical studies when consumed with pasta to show benefits such as reduced GI [100] and reduced blood pressure [107] while only Kamble et al. (2019) [79] study using soy-okra provided evidence of a reduction in in vitro GI. Wholemeal pasta has been shown to have a low GI from a survey by Di Pede et al. (2021) [102] ranging from 35–65, while studies examining the in vitro starch digestion of wholemeal pasta are rare. One study found no effect on the in vitro starch digestion in pasta prepared with fine bran 10–30% [27]. Vegetables have been added to pasta for many years with two studies showing low pasta GI with added vegetable pulps [131] supported by the in vitro studies [77,79]. The other ingredients added to pasta listed in Table 2 seem not to have been evaluated in human clinical trials (gums, debranning fractions, wheat embryo, herbs, protein extracts) so more evaluation is needed. As always, cost of human trials can be prohibitive as well as obtaining ethics approval. Also, there is a need to look at other health indicators besides those discussed in this review such as ingredients added to pasta that can demonstrate benefits to mental health, slowing aging, improving the microbiome health.

Overall, it seems many of the in vitro claims are met by the in vivo results although the level of affect in vivo could be higher than the in vitro studies suggest. For example, raising the amylose content of pasta to mid-40s% while significantly increasing the in vitro starch digestion extent had no significant impact on the GI in a 10 subject glucose tolerance test [7]. The in vitro studies provide a guide to the likely impact in the human but claims for health benefits require proof from the human feeding trials and longer-term data to provide good evidence for using a functional pasta for health benefits. Much more research is needed in this area. Interactions between active compounds and protein matrix while understood for some ingredients like bran, inulin, soluble fibre and resistant starch [6] are not understood for many novel approaches proposed in Table 1. Only a few studies have considered synergism or interactions between individual compounds affecting pasta product quality. While a health benefit is sought after in the many studies discussed, a very important consideration is consumer acceptance of the functional pasta. Many of the studies listed in Table 2 evaluate the technological quality of the resultant pasta with a range of instrumental, cooking procedures and colour as well as the important sensory analysis, most often using a trained panel mostly limited to 10 people. More expensive and time-consuming consumer panels involving many people are needed to give an indication of the market acceptance of the product since taste, appearance, smell and texture are important to consumers. However, gender, race, country of origin can affect peoples perceived acceptance of functional foods [132]. Generally, these studies are not done for specific functional foods but more generally for food categories like wholegrain foods [132]. Another important consideration in the manufacture and consumption of functional pasta is the storage stability. Dried pasta made from 100% semolina typically has a water activity in the range 0.3–0.5 [133] and if stored in sealed containers typically lasts at least 2 or more years, often well passed the shelf-life given by the manufacturer, which is often conservative. However, the composition of the pasta with added ingredients needs to be considered. For example, any egg products used in the manufacture of the dried pasta may not be as stable for such a length because of their high content of lipids, the pasta can turn rancid. Discolouration or off-odours are good indicators of spoilage. Fresh pasta has a much shorter shelf life of 2–3 days with refrigeration because of the high water activity (0.92–0.99) [133] and will deteriorate rapidly if not stored properly. Various chemicals and natural antimicrobials can be used to extend shelf-life [6]. There are limited studies on the storage stability of pasta and impacts on pasta nutritional value. One example is the use of modified packaging using high-density polyethylene and biaxially oriented polypropylene films were compared with the former providing a longer shelf life for multigrain pasta [134]. Another investigated lipid oxidation in spaghetti enriched in long chain *n*-3 polyunsaturated fatty acids with functional spaghetti having a shelf life comparable to control pasta [135]. Thus, it is important to include storage stability studies in functional food design to ensure no deterioration in the functional value occurs with storage.

## 3. Conclusions

Pasta is a popular food and has already been shown to be a good method to incorporate increased nutritional or functional compounds. Care is needed to ensure good technological quality in pasta with a substituted ingredient and consumer acceptability at an affordable price. Therefore, the manufacturer of such products must be profitable, and a ready supply of the desired ingredient be assured before a manufacturer prepares such products as well as a likely market. Interactions between active compounds and protein matrix while understood for some ingredients like bran, inulin, soluble fibre and resistant starch, are not understood for many novel approaches proposed in Table 1. Only a few studies have considered synergism or interactions between individual compounds affecting pasta product quality. However, legislation in many countries require proof before a health claim can be made on a food, such as low GI, cholesterol lowering, heart safe etc. For this reason, more research is needed to evaluate the most promising functional pasta with human clinical trials to validate the actual health benefit. Health claims together with good taste, texture and appearance at an acceptable price will help drive consumer demand for such foods.

## Data Availability

The data presented in this study are available on request from the corresponding author. The data are not publicly available due to institutional privacy requirements.

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
