# Peer review of "Development of Novel Pasta Products with Evidence Based Impacts on Health—A Review"

_foods, 2022, doi:10.3390/foods11010123_

Round 1

Reviewer 1 Report

Dear Mike Sissons,

The manuscript under review deals an interesting subject, since it provides an overview of pasta made from durum wheat where the semolina is substituted in part with a range of ingredients in order to make pasta with a functional benefit.

The document is well written, and the subject of the manuscript is interesting. The paper will contribute to the scientific advancement of the relevant research area. However, there are some points to be considered:

  • It would be interesting that author provides a graphical abstract
  • In the text, reference numbers should be placed in square brackets [ ]. It must be carefully corrected in the manuscript because of references don´t follow the required format. Please, check it.

Other specific considerations are as follows:

Abstract section:

  • Line 12: Please, change “and overview” by “an overview”.

Introduction section:

  • Line 22: Please, change “with cereals” by “being cereals”.
  • Line 28: Please, change “T aestivum” by “ aestivum”.
  • Line31: Please, insert comma (,) between “pasta” and “which”.
  • Line 32: It is wrong to affirm that pasta lacks dietary fiber. Please, modify this sentence.
  • Line 51: I recommend include in the text a cite that justify the affirmation “These diseases represent the main non-communicable cause of death”.
  • Line 55: I recommend include in the text the cited reference in square brackets [] and placed the web address in reference section in the appropriate format.
  • Line 59: I suggest including a cite at the end of this paragraph.
  • Line 59: I suggest including a cite in this paragraph.
  • Lines 75 – 79: I am not in agree when author says that plant breeding programs are not focused on nutrition neither health benefits aspects. I recommend the review of publications derived from HEALTHGRAIN program.
  • Line 90. It is not appropriate to cite Table 3 before than Table 2. Please, check it.
  • Line 106: I suggest including a cite at the end of this paragraph.

Methods section:

  • Line 124: The number section must be changed (2. Methods).
  • Line 127: Please, cite the references using the required format.
  • Table 1. Please, include a column with corresponding references.

Results and Discussion section:

  • Line 166: I suggest including a cite at the end of this paragraph.
  • Line 176: The author cites a study performed in 1983. However, in lines 115-116, it is indicated that “The present review aims to provide an overview of recently published scientific articles from the year 2000 to 2021…” Please, check it.
  • Line 190: I suggest change “other additives to pasta” by “other ingredients added to pasta”.
  • Line 195: Please, insert “that” between “noted” and “their”.
  • Line 198: Please, correct the authors name corresponding to reference nº 29 (the adequate form is: Goñi and Valentín-Gamazo).
  • Line 214: Which are the proposed health claims?
  • Line 218: Bacillus coagulans is a probiotic, but not prebiotic. Please, check it.
  • Line 241: The author cites a study performed in 1999. However, in lines 115-116, it is indicated that “The present review aims to provide an overview of recently published scientific articles from the year 2000 to 2021…” Please, check it.
  • Line 248: The author cites a study performed in 1996. However, in lines 115-116, it is indicated that “The present review aims to provide an overview of recently published scientific articles from the year 2000 to 2021…” Please, check it.
  • Line 250: Please, change “28d” by “28 days”.
  • Line 255: Please, delete one point (.) after the word “study”.
  • Line 267: I recommend entitled this subsection as “Hypocholesterolemic properties and benefit effects on cardiovascular disease (CVD)”
  • Line 321: Please, cite the references using the required format.
  • Line 342: Please, change “30d” by “30 days”.
  • Line 343-347: I suggest move this subsection and include it at the end of the subsection namely “Oxidative stress effects” under the common tittle “Oxidative stress and aging effects”
  • Line 348 – 362: I recommend place this subsection after “Hypocholesterolemic effects and CVD”.
  • Line 355: Please, change “assessedby30” by “assessed by 30”.
  • Line 392: Please, review the sentence “After consumption of wholemeal pasta, blood? Glucose and triglyceride” and correct it.
  • Line 396: Please, delete a point (.).
  • Line 405-406: Please, cite the references using the required format.
  • Line 408: Please, change “additives” by “components” or “ingredients”.
  • Line 420-421: Please, cite the references using the required format.
  • Line 432: Please, cite the references using the required format (29, 75).
  • Line 444: Please, cite the references using the required format (79, 82).
  • Table 3: All sentences must begin by capital letter. Please, check it.
  • Table 3: In column namely “substitution (%)”, I suggest indicating only the corresponding range (p.e. 0 – 20).
  • Table 3: I suggest change the tittle “Dietary fibres” by “Other dietary fibres components”.
  • Table 3. I recommend change the tittle “Fish products” by “Fish products and algae”.
  • Table 3: In section namely “Inulin addition”, Why has author included “Pastas with elderberry juice Concentrate (EJC) and Hi-maize starch or apple pectin” here?

References’ section:

  • Line 552: Please, review this cite (Goñi and Valentín-Gamazo, 2003).

I consider that the manuscript titled “Development of Novel Pasta Products with Evidence based Impacts on Health, a Review” needs major revision.

Author Response

The manuscript under review deals an interesting subject, since it provides an overview of pasta made from durum wheat where the semolina is substituted in part with a range of ingredients in order to make pasta with a functional benefit.

The document is well written, and the subject of the manuscript is interesting. The paper will contribute to the scientific advancement of the relevant research area. However, there are some points to be considered:

I would like to thank the reviewer for their meticulous review and helpful suggestions,

  • It would be interesting that author provides a graphical abstract
  • A graphical abstract has been included
  • In the text, reference numbers should be placed in square brackets [ ]. It must be carefully corrected in the manuscript because of references don´t follow the required format. Please, check it.
  • Correct brackets used throughout

Other specific considerations are as follows:

Abstract section:

  • Line 12: Please, change “and overview” by “an overview”.
  • Corrected, L10

Introduction section:

  • Line 22: Please, change “with cereals” by “being cereals”.
  • Changed to “with cereals, tubers and pulses being the main dietary sources” L22
  • Line 28: Please, change “T aestivum” by “aestivum”.
  • Corrected L26
  • Line31: Please, insert comma (,) between “pasta” and “which”.
  • Added L29
  • Line 32: It is wrong to affirm that pasta lacks dietary fiber. Please, modify this sentence.
  • Text now modified L30
  • Line 51: I recommend include in the text a cite that justify the affirmation “These diseases represent the main non-communicable cause of death”.
  • Ref 4 is suitable text has been modified.
  • Line 55: I recommend include in the text the cited reference in square brackets [] and placed the web address in reference section in the appropriate format.
  • I have followed the advice and made the change
  • Line 59: I suggest including a cite at the end of this paragraph.
  • Two references have been included L57
  • Line 59: I suggest including a cite in this paragraph.
  • A reference has been added L60
  • Lines 75 – 79: I am not in agree when author says that plant breeding programs are not focused on nutrition neither health benefits aspects. I recommend the review of publications derived from HEALTHGRAIN program.
  • A reference has been added and the wording changed to reflect the work of the HEALTHGRAIN project. L73-75
  • Line 90. It is not appropriate to cite Table 3 before than Table 2. Please, check it.
  • Table 2 and 3 have been renumbered where 2 now becomes 3 although its position is before table 2 because it is a smaller table. Editor can advise if this is acceptable.
  • Line 106: I suggest including a cite at the end of this paragraph.
  • A new reference has been added which talks about bioavailability L105

Methods section:

  • Line 124: The number section must be changed (2. Methods).
  • I am unsure what is being requested, Editor please advise
  • Line 127: Please, cite the references using the required format.
  • Done
  • Table 1. Please, include a column with corresponding references.
  • References now included

Results and Discussion section:

  • Line 166: I suggest including a cite at the end of this paragraph.
  • In the submitted manuscript line 166 refers to “durum wheat pasta made from a high amylose durum wheat (with elevated resistant starch), at least.” Which is reference [7] so I am unsure what text the reviewer is referring to. Editor please advise. L158-161 represents my opinion and so a ref is not needed.
  • Line 176: The author cites a study performed in 1983. However, in lines 115-116, it is indicated that “The present review aims to provide an overview of recently published scientific articles from the year 2000 to 2021…” Please, check it.
  • I have added this statement “However, where appropriate relevant pre- 2000 references are included.” As in some cases it is appropriate to use some older references for the discussion. L110
  • Line 190: I suggest change “other additives to pasta” by “other ingredients added to pasta”.
  • I note that this text (“other additives to pasta”) in original submitted word file is found at line 174 not 190. Replacement has been made. L187
  • Line 195: Please, insert “that” between “noted” and “their”.
  • Done L192
  • Line 198: Please, correct the authors name corresponding to reference nº 29 (the adequate form is: Goñi and Valentín-Gamazo).
  • Done L195
  • Line 214: Which are the proposed health claims?
  • Text has been modified L209
  • Line 218: Bacillus coagulans is a probiotic, but not prebiotic. Please, check it.
  • Corrected to probiotic L213
  • Line 241: The author cites a study performed in 1999. However, in lines 115-116, it is indicated that “The present review aims to provide an overview of recently published scientific articles from the year 2000 to 2021…” Please, check it.
  • Have revised the statement in lines 115-116 allowing use of references older than 2000.
  • Line 248: The author cites a study performed in 1996. However, in lines 115-116, it is indicated that “The present review aims to provide an overview of recently published scientific articles from the year 2000 to 2021…” Please, check it.
  • Have revised the statement in lines 115-116 allowing use of references older than 2000.
  • Line 250: Please, change “28d” by “28 days”.
  • Changed L242
  • Line 255: Please, delete one point (.) after the word “study”.
  • Removed L246
  • Line 267: I recommend entitled this subsection as “Hypocholesterolemic properties and benefit effects on cardiovascular disease (CVD)”
  • Revised to Hypocholesterolemic properties and beneficial effects on cardiovascular disease (CVD)
  • Line 321: Please, cite the references using the required format.
  • done
  • Line 342: Please, change “30d” by “30 days”.
  • Done L344
  • Line 343-347: I suggest move this subsection and include it at the end of the subsection namely “Oxidative stress effects” under the common tittle “Oxidative stress and aging effects”
  • done
  • Line 348 – 362: I recommend place this subsection after “Hypocholesterolemic effects and CVD”.
  • done
  • Line 355: Please, change “assessedby30” by “assessed by 30”.
  • Done L312
  • Line 392: Please, review the sentence “After consumption of wholemeal pasta, blood? Glucose and triglyceride” and correct it.
  • Revised L397
  • Line 396: Please, delete a point (.).
  • done
  • Line 405-406: Please, cite the references using the required format.
  • done
  • Line 408: Please, change “additives” by “components” or “ingredients”.
  • Done L412
  • Line 420-421: Please, cite the references using the required format.
  • done
  • Line 432: Please, cite the references using the required format (29, 75).
  • done
  • Line 444: Please, cite the references using the required format (79, 82).
  • done
  • Table 3: All sentences must begin by capital letter. Please, check it.
  • All tables conform to formatting
  • Table 3: In column namely “substitution (%)”, I suggest indicating only the corresponding range (p.e. 0 – 20).
  • done
  • Table 3: I suggest change the tittle “Dietary fibres” by “Other dietary fibres components”.
  • done
  • Table 3. I recommend change the tittle “Fish products” by “Fish products and algae”.
  • done
  • Table 3: In section namely “Inulin addition”, Why has author included “Pastas with elderberry juice Concentrate (EJC) and Hi-maize starch or apple pectin” here?
  • Added to resistant starch heading

References’ section:

  • Line 552: Please, review this cite (Goñi and Valentín-Gamazo, 2003).

revised

I consider that the manuscript titled “Development of Novel Pasta Products with Evidence based Impacts on Health, a Review” needs major revision.

Reviewer 2 Report

too few keywords .. can add functional noodles, functional food 

List of abbreviations - of course they are very important and crucial for the free navigation of the review article ... but should they be in this place? 

In addition, I believe that the use of abbreviations when describing technological processes or the properties of pasta makes it difficult to read freely. Descriptions of additives are also not obvious - prickly pear extract. it is indeed a large part of the obvious abbreviations RS- starch abundant and it is easy to find in the memory when reading. I believe that the following shortcuts should not be introduced at work:

  • pasta making quality
  • Non starch polysaccharides
  • Optimum cooking time
  • Cooking loss
  • Water absorption
  • locust bean gum
  • Amaranth seed flour
  • Dried amaranth leaves
  • Lupin protein isolate
  • Opuntia cladode extract

Section of the method .. should be titled slightly differently. In this form, it indicates that research has been performed and not reviewed. this is a review work..so the methods are wrongly introduced. Maybe the title "Review of methods" would be better.

Table 1 is not readable, it is necessary to introduce a clear separation of subsequent content so that the description in the lines corresponds to the columns. I know that this is the publisher's requirement as to the form of the table, but maybe a clearer separation of the EP will improve the current form.

Completely and multidirectionally collected and described impact of various enriched pasta on human health.

A valuable overview of pasta as a new functional food product. It would be valuable to indicate how the taste qualities of these pastas are assessed, as the addition of functional ingredients was sometimes of a large share. For most consumers, apart from being aware of a healthy product, there is also its sensory effect - taste.

According to my assessment, the review work submitted for evaluation lacked two issues, the addition of which would increase the value of the work even more.
The first one is pasta obtained by extrusion, which does not require culinary processing and their glycemic index is quite high, which indicates that not only the composition but also the method of obtaining the pasta shape its properties.

The second aspect - as you know, most pasta is dried after pressing to a safe level of moisture, which allows for quite a long-term storage. It would be important to add some literature reports on changes in nutritional value, health and safety during storage.

Author Response

Comments and Suggestions for Authors

too few keywords .. can add functional noodles, functional food 

Added but prefer functional pasta to functional noodles since the former is the focus of this review

List of abbreviations - of course they are very important and crucial for the free navigation of the review article ... but should they be in this place? 

Editor please advise

In addition, I believe that the use of abbreviations when describing technological processes or the properties of pasta makes it difficult to read freely. Descriptions of additives are also not obvious - prickly pear extract. it is indeed a large part of the obvious abbreviations RS- starch abundant and it is easy to find in the memory when reading. I believe that the following shortcuts should not be introduced at work:

  • pasta making quality
  • Non starch polysaccharides
  • Optimum cooking time
  • Cooking loss
  • Water absorption
  • locust bean gum
  • Amaranth seed flour
  • Dried amaranth leaves
  • Lupin protein isolate
  • Opuntia cladode extract

All removed from abbreviation list and spelled out in full

Section of the method .. should be titled slightly differently. In this form, it indicates that research has been performed and not reviewed. this is a review work..so the methods are wrongly introduced. Maybe the title "Review of methods" would be better.

Changed thanks for the suggestion. I did have issues with the conventional term “Methods” I have used your suggestion unless the editor wishes to change.

Table 1 is not readable, it is necessary to introduce a clear separation of subsequent content so that the description in the lines corresponds to the columns. I know that this is the publisher's requirement as to the form of the table, but maybe a clearer separation of the EP will improve the current form.

Thanks for the suggestion, yes table can be difficult to read so have re-formatted and hopefully improved clarity.

Completely and multidirectionally collected and described impact of various enriched pasta on human health.

 I am unsure if any action is required and don’t understand the statement.

A valuable overview of pasta as a new functional food product. It would be valuable to indicate how the taste qualities of these pastas are assessed, as the addition of functional ingredients was sometimes of a large share. For most consumers, apart from being aware of a healthy product, there is also its sensory effect - taste.

 This has been added :While a health benefit is sought after in the many studies discussed a very important consideration is consumer acceptance of the functional pasta. Many of the studies listed in Table 2 evaluate the technological quality of the resultant pasta with a range of instrumental, cooking procedures and colour as well as the important sensory analysis, most often using a trained panel mostly limited to 10 people. More expensive and time consuming consumer panels involving many people are needed to give an indication of the market acceptance of the product since taste, appearance, smell and texture are important to consumers. However, gender, race, country of origin can affect peoples perceived acceptance of functional foods [136]. Generally these studies are not done for specific functional foods but more generally for food categories like wholegrain foods [136]. Lines 477-485

According to my assessment, the review work submitted for evaluation lacked two issues, the addition of which would increase the value of the work even more.
The first one is pasta obtained by extrusion, which does not require culinary processing and their glycemic index is quite high, which indicates that not only the composition but also the method of obtaining the pasta shape its properties.

The GI of extruded pasta (the most common method for pasta preparation) has variable GI ranging from 18-93 as measured by in vivo test (Di Pede et al 2021) and this is mentioned therefore the comment that extruded pasta is high GI is not strictly correct. While the Di Pede paper did not provide details on how the pasta samples were prepared, most likely they were by extrusion method. It is true that besides composition other factors affect the measurement of GI including the number of subjects, their racial mix and sex as well as the form the pasta takes such as its shape, diameter, and how it was processed (extrusion conditions, drying conditions, water:solid ratio etc). I do not feel a discussion of details affecting GI in pasta is the subject of this review as there are many articles looking at such factors. However, I have provided a reference to cover this area for readers interested in more detail on this topic of pasta processing method impact on starch digestion at L256-57.

The second aspect - as you know, most pasta is dried after pressing to a safe level of moisture, which allows for quite a long-term storage. It would be important to add some literature reports on changes in nutritional value, health and safety during storage.

I have included some information and references on this topic at Line 467-501.

Submission Date

29 November 2021

Date of this review

07 Dec 2021 18:12:08

Round 2

Author Response

There are no comments from reviewer 1 after revision